# Biocompatible Alginate Hydrogel Film Containing Acetic Acid Manifests Broad-Spectrum Antiviral and Anticancer Activities

**DOI:** 10.3390/biomedicines11092549

**Published:** 2023-09-16

**Authors:** Alba Cano-Vicent, Alberto Tuñón-Molina, Hamid Bakshi, Iman M. Alfagih, Murtaza M. Tambuwala, Ángel Serrano-Aroca

**Affiliations:** 1Biomaterials and Bioengineering Lab, Centro de Investigación Traslacional San Alberto Magno, Universidad Católica de Valencia San Vicente Mártir, 46001 Valencia, Spain; alba.cano@mail.ucv.es (A.C.-V.); alberto.tunon@ucv.es (A.T.-M.); 2Hormel Institute, University of Minnesota, Austin, MN 55912, USA; hamid.bakshi@gmail.com; 3Department of Pharmaceutics, College of Pharmacy, King Saud University, Riyadh 4545, Saudi Arabia; fagih@ksu.edu.sa; 4Brayford Pool Campus, Lincoln Medical School, University of Lincoln, Lincoln LN6 7TS, UK

**Keywords:** acetic acid, alginate, hydrogel, antiviral, anticancer, toxicity, industrial pharmacy

## Abstract

Acetic acid, a colourless liquid organic acid with a characteristic acrid smell, is obtained naturally and has applications in both the food and pharmaceutical industries. It has been reported to have beneficial uses for lifestyle-related diseases, and its efficient disinfectant properties are well known. In this study, an alginate crosslinked with Ca^2+^ hydrogel film was treated with acetic acid to explore its biological properties for biomedicine. The results showed that the novel calcium alginate/acetic acid film was biocompatible in vitro using human keratinocyte cells and in vivo with *Caenorhabditis elegans*. It also had antiviral properties against enveloped and non-enveloped viruses and anticancer properties against melanoma and colon cancer cells. This novel film thus showed promise for the biomedical and pharmaceutical industries, with applications for fabricating broad-spectrum antiviral and anticancer materials.

## 1. Introduction

Alginate is a biopolymer obtained mainly from brown algae [1]. They are linear polymers formed by alternating guluronic (α-L-guluronic) and mannuronic (β-D-mannuronic) acid units, polyguluronic zones, and polymannuronic zones. This natural polymer has a large number of industrial and biomedical applications such as water treatment, tissue engineering, and drug delivery [2]. One of the most common ways to use alginate is crosslinked with divalent cations of Ca^2+^ in the form of a hydrogel [3]. Hydrogels are three-dimensional crosslinked polymeric structures that can absorb large amounts of water, due to the hydrophilic groups of their polymeric skeleton [4]. They also have high strength, good flexibility, and are insoluble in water. Alginate hydrogels can be prepared with no toxicity in both in vitro or in vivo assays [5,6]. Their biological properties can be enhanced by adding other compounds with intrinsic antimicrobial properties, such as silver, zinc, cobalt, copper, or carbon-based materials [7,8].

Acetic acid (Ac), a clear colourless liquid organic acid with a characteristic stinging odour, is obtained naturally [9,10]. The group of aerobic bacteria known as acetic acid bacteria (AAB) is able to oxidise ethanol to acetic acid [11], an important process in the food industry because it occurs in the fermentation process and can convert some types of sugars and alcohols into organic acids [12]. AAB can be isolated from different types of fruit, flowers, and fermented foods [13].

The process of obtaining acetic acid originated around 2000 BC to make products such as vinegar [14]. In addition to its use in the food industry, this product is beneficial for lifestyle-related diseases such as hypertension, hyperlipidemia, and obesity [15]. Since the times of Ancient Greece, vinegar has been used as a disinfectant [16], as has been confirmed in many studies [17,18]. Sloss et al. showed that Ac eliminated *Pseudomonas aeruginosa* after two weeks of treatment, a common bacterium in infections [19]. The bactericidal effect on Gram-positive organisms, such as *Staphylococcus aureus* and *Enterococcus* spp., or Gram-negative bacteria, like *P. vulgaris* and *A. baumannii,* have also been shown [20]. Ac is an effective and rapid tuberculocidal disinfectant [21], whereas some studies have shown the antiviral [22,23,24] and anticancer activity [25,26,27,28,29] of its different derivates. Recent research has found that acetic acid prevents SARS-CoV-2 from replicating [30] using transmission electron microscopy to show how this acid alters the structure of the SARS-CoV-2 virion. It interferes with the SARS-CoV-2 spike the protein’s ability to bind to ACE2, the main SARS-CoV-2 cell receptor.

Ac’s antibacterial, antiviral, and anticancer properties thus make it a perfect candidate for biomedical use. In this study, an alginate crosslinked with Ca^2+^ hydrogel films was prepared by solvent casting [31] and subsequent treatment with acetic acid. The aim was to study the biological properties of this novel hydrophilic film in terms of toxicity, antiviral activity, and anticancer properties.

## 2. Material and Methods

### 2.1. Materials

Sodium alginate and calcium chloride (≥93.0%) were purchased from Sigma-Aldrich (Saint Louis, MO, USA). The alginate used had previously been characterized [1] (FG = 0.463, FM = 0.537, or an M/G ratio of 1.16, FGG = 0.282, FMG = 0.181, FMM = 0.356, FGGM = FMGG = 0.048, and FGGG = 0.234). The guluronic acid blocks were mostly arranged in short G-blocks (average length~7). Acetic acid (≥99.8%) from Honeywell, Sigma-Aldrich (Saint Louis, MO, USA). Foetal bovine serum (FBS), DMEM low glucose, penicillin-streptomycin (P/S), L-glutamine, and Epidermal Growth Factor (EGF) were obtained from Life Technologies (Gibco, Karlsruhe, Germany). Bacteriological agar was obtained from Scharlau (Ferrosa, Barcelona, Spain). Tryptic soy broth (TSB) and tryptic soy agar (TSA) were purchased from Liofilchem (Teramo, Italy).

### 2.2. Formulation of Hydrogel Film

In total, 0.25 g of sodium alginate was dissolved in 30 mL of distilled water by magnetic stirring for 1 h at 24 ± 0.5 °C. The solution was placed in a Petri dish and left, first at room temperature (24 ± 0.5 °C) for 24 h and then at 37 °C in an oven for 48 h. To crosslink the alginate, 500 mL of 1% *w*/*v* (5 g of calcium chloride in 500 mL of distilled water) calcium chloride solution was prepared. A small amount of this solution was put on the sodium alginate film and, after separation from the Petri dish, was placed in the calcium chloride solution for 2 h at 24 ± 0.5 °C. To remove any residual crosslinker, the film was washed three times in a water solution, poured into a Petri dish and left, first at room temperature (24 ± 0.5 °C) for 24 h and then at 37 °C in an oven for 48 h. Acetic acid was diluted in distilled water to obtain a 10% *v*/*v* solution. A total of 10 mL of 10% *v*/*v* of dilute acetic acid was put in the alginate crosslinked with calcium film. The film treated with acetic acid (Ac10) was then poured into a Petri dish and left, first at room temperature (24 ± 0.5 °C) for 24 h and then at 37 °C in an oven for 48 h. Another alginate crosslinked with calcium film was not treated with acetic acid (Ac0) and was used as the control sample in the assays. The amount of acetic acid introduced into the calcium alginate film was determined gravimetrically (*n* = 6). 10 mm diameter discs were obtained from the films by means of a cylindrical punch (Figure 1).

Figure 1 shows that both films presented similar macroscopic morphologies. These samples were then subjected to ultraviolet radiation for 1 h per side for sterilisation.

Similar calcium alginate films without acetic acid (control film) have been deeply characterized by Fourier transform infrared spectroscopy, field emission scanning electron microscope with energy dispersive X-ray spectrometry, water sorption, and degradation studies [2,32].

### 2.3. Water Absorption Test

Water absorption at different times (30 min, 5 h, and 24 h) of the dried films (at 60 °C for 48 h) were performed following a previously described protocol by our research group [32]. The films were dried at 60 °C for 48 h to constant weight, after which they were weighed and placed in 100 mL of distilled water and left in an oven at 37 °C. It was thus weighed at different water sorption times (30 min, 5 h, and 24 h) (*h*), calculated by Equation (1):
(1)
$$h=\left(\frac{m_{hydrated\;film}-m_{dry\;film}}{m_{dry\;film}}\right)$$

where *m_hydrated film_* is the weight of the swollen film for each time, and *m_dry film_* is the weight of the dry film.

### 2.4. Toxicological Study

The discs were placed in a 6-well plate with DMEM (Biowest SAS, Nuaillé, France) without FBS, at a volume ratio of 3 cm^2^/mL, according to ISO-10993, which recommends this rate for tube wall, slab, and small moulded articles followed in the toxicological study, proliferation assay, and anticancer study.

A culture medium was used for growth based on DMEM low glucose, supplemented with FBS 10%, 1% *w*/*v* penicillin (Lonza, Verviers, Belgium), and 1% *w*/*v* streptomycin (HyClone, GE Healthcare Life Sciences, Piscataway, NJ, USA). Human keratinocytes were seeded into 96-well plates at a density of 10^4^ cells/well and grown 24 h at 37 °C in a 5% CO_2_ humidified atmosphere. The medium was then replaced by 100 μL with the corresponding extractions of each sample and incubated for 24 h in the same conditions. Six replicate samples were prepared in wells for each concentration, plus an untreated control group. Cytotoxicity was evaluated by the 3-(4,5-dimethylthiazol-2-yl)-2,5-diphenyl tetrazolium (MTT) assay. MTT was added to replace the cells’ medium and incubated for 2 h in the same conditions as the culture. Formazan crystals were then solubilised with DMSO, and cell viability was determined from the absorbance at 550 nm on a Varioskan micro plate reader (ThermoScientific, Mississauga, ON, Canada).

### 2.5. Proliferation Assay

For the study of proliferative activity [33], the cells were seeded in 96-well plates but at a density of 5 × 10^3^ cells/well. In this case, the culture medium to grow the cells included 0.5% FBS. The cells were cultured for 72 h, after which the MTT assay was carried out on cell growth. A positive proliferation control was also included for each time, exposed to epidermal growth factor (EGF) at a concentration of 15 ng/mL. The tests were carried out in sextuplicate.

### 2.6. Anticancer Study

Colon cancer cell line (HT-29) and melanoma (B16) [34,35,36,37] were used for the anticancer assay. The culture medium used to grow the cells was based on DMEM with 10% FBS, 1% L-glutamine and 1% penicillin/streptomycin (Thermo Scientific Hyclone, Logan, UT, USA). The cells were seeded in 96-well plates at a density of 10^5^ cells/well. The cells were cultured for 24 h, after which the MTS (3-(4,5-dimethylthiazol-2-yl)-5-(3-carboxymethoxyphenyl)-2-(4-sulfonyl)-2H-tetrazolium) growth assay was carried out.

### 2.7. In Vivo Toxicity Tests

The in vivo toxicity (acute and chronic) in terms of survival rate, reproduction, and body length was tested using the *Caenorhabditis elegans* model according to a protocol previously described in detail by our group [32]. The *Caenorhabditis elegans* model was used to study in vivo toxicity. Nematodes were provided by the Caenorhabditis Genetics Center (CGC, Minneapolis, MN, USA). An N2 strain was maintained and propagated on nematode growth medium (NGM) with OP50 *E. coli* at 25 °C for the experiments. The worms were synchronised by washing the plates with the nematodes in 5 mL of distilled water. The tubes with the worms were centrifuged at 1300 r.p.m. for 3 min, and the pellet was resuspended in 100 µL of distilled water and 700 µL of a 5% bleaching solution, after which the mixture was vortexed every 2 min. This step was repeated 5 times. The tubes were centrifuged at 700× *g* for 3 min, and the pellet with the worm eggs was resuspended in 800 µL of distilled water. This step was carried out three times. In the last centrifugation, the pellet was resuspended in 100 µL of distilled water to transfer it to an NGM plate with OP50 *E. coli*. The plate with the nematode eggs was incubated for 72 h at 25 °C to obtain worms in the L1 stage population. After 72 h, the NGM plates were washed and centrifuged with the same procedure, and the pellet was resuspended in 3 mL of potassium medium.

The samples were extracted by the same method used for in vitro toxicity, a potassium medium (2.36 g potassium chloride, 3 g sodium chloride in 1 L distilled water, autoclaved) and incubated for 72 h at 25 °C.

For the assay, a mixture with 62.5 µL of a 1:250 suspension of cholesterol (5 mg/mL in ethyl alcohol) in sterile potassium medium, 62.5 µL of a 50× concentrated OP50 *E. coli* culture, 115 µL of potassium medium, and 250 µL of the pertinent extract were placed in a 48-well plate, and 50–100 worms were added to each well. A positive control (worms incubated only with medium without extracts) and a negative control (worms incubated with a toxic zinc dilution) were performed. The plates were stamped with parafilm and placed in an orbital shaker at 25 °C and 120 r.p.m. for 24 and 72 h. The mixture incubation with the extracts and the worms was divided into 10 drops of 50 µL and placed under a microscope (Motic BA410E including Moticam 580 5.0 MP) for the survival rate of *C. elegans*. The numbers of living and dead worms were counted. To analyse reproduction, three worms were placed in a new OP50-seeded NGM plate and incubated for 48 h at 25 °C to count the eggs under a microscope. Body length was measured in a photo taken under the microscope by Motic Images Plus 3.0 software. Six independent replicates (*n* = 6) were conducted.

### 2.8. Double-Stranded RNA Extraction and Quantification

Double-stranded RNA extraction and quantification of both types of viruses (bacteriophage phi6 and bacteriophage MS2) were performed to make sure that the viral particles did not remain attached to the Ac0 and Ac10 films before the antiviral tests to avoid false results. RNA was extracted following the RNA extraction protocol provided by the Norgen Biotek Corp. (Thorold, ON, Canada) [37]. These RNA extraction and quantification tests were performed according to a protocol previously described in detail by our research group [32].

### 2.9. Antiviral Test

#### 2.9.1. Using Enveloped Bacteriophage Φ6

The Gram-negative *Pseudomonas syringae* (DSM 21482) and the enveloped bacteriophage Φ6 (DSM 21518) were obtained from the Leibniz Institute DSMZ-German Collection of Microorganisms and Cell Cultures GmbH (Braunschweig, Germany). The bacteria were grown on a TSA plate and in liquid TSB at 25 °C at a speed of 120 rpm. The bacteriophage was propagated following the specifications provided by the Leibniz Institute.

The infective activity of the bacteriophage was determined by the double-layer method [37] A titre of approximately 1 × 10^6^ PFU/mL of a bacteriophage suspension in TSB, 50 μL, was added to each sample and incubated for 5 min, 30 min, 5 h, and 24 h, after which the discs with the bacteriophage suspension were placed in a falcon tube with 10 mL of TSB, sonicated for 5 min at 25 °C, and subsequently vortexed for 1 min. Serial dilutions were made for each sample. A total of 100 μL of the host strain at OD_600nm_ = 0.5 was mixed with 100 μL of the bacteriophage dilutions. This suspension was mixed with 4 mL of top agar (TSB + 0.75% bacteriological agar) with 1 mM CaCl_2_ to be finally placed on TSA plates, which were incubated at 25 °C for 24 h. The bacteriophage titre of each sample was calculated and expressed in PFU/mL for comparison with the control, which consisted of the bacteriophage–bacteria suspension that had not been in contact with any disc. The control also ensured that the sonication/vortexing treatment did not affect the bacteriophage’s infectious activity. The antiviral tests were performed in triplicate on two different days (*n* = 6) to guarantee reproducible results.

#### 2.9.2. Using Non-Enveloped Bacteriophage MS2

*E. coli* (DSM 5695) and bacteriophage MS2 (DSM 13767) were obtained from the Leibniz Institute DSMZ-German Collection of Microorganisms and Cell Cultures GmbH (Braunschweig, Germany). The bacteria were grown on a TSA plate and then in liquid TSB at 37 °C at a speed of 240 rpm. The bacteriophage was propagated following the specifications provided by the Leibniz Institute.

The bacteriophage’s infective activity was determined by the double-layer method [37]. A total of 50 μL of a titre of approximately 1 × 10^6^ PFU/mL of a bacteriophage suspension in TSB was added to each sample and incubated for 5 min, 30 min, 5 h, and 24 h. The discs with the bacteriophage suspension were placed in a falcon tube with 10 mL of TSB and sonicated for 5 min at 37 °C and subsequently vortexed for 1 min. Serial dilutions were made for each sample. A total of 100 μL of the host strain at OD_600nm_ = 0.2 was mixed with 100 μL of each bacteriophage dilution. This suspension was mixed with 4 mL of top agar (TSB + 0.75% bacteriological agar) with 1 mM CaCl_2_ to be finally placed on TSA plates, which were incubated at 37 °C for 24 h. The bacteriophage titre of each sample was calculated and expressed in PFU/mL for comparison with control, which consisted of the bacteriophage–bacteria suspension that had not been in contact with any disc. The control also ensured that the sonication/vortexing treatment did not affect the bacteriophage’s infectious activity. The antiviral tests were performed in triplicate on two different days (*n* = 6) to guarantee reproducible results.

## 3. Results and Discussion

### 3.1. Film Composition

The percentage weight of acetic acid was determined by gravimetric analysis for each sample film (Table 1).

The goal of this study was to produce an antiviral and anticancer alginate-based film with the addition of this low amount of acetic acid (15.95 ± 1.45% Weight) without compromising the in vivo and in vitro biocompatibility of the biopolymer.

### 3.2. Water Absorption

After drying the films at 60 °C for 48 h to constant weight, they were placed in 100 mL of distilled water in an oven at body temperature (37 °C). The films were weighed at three different times (30 min, 5 h, and 24 h) to calculate the h absorption, defined as the mass of water divided by the mass of dry film (Figure 2).

The h absorption of liquid water at 37 °C showed no statistically significant differences between untreated film (Ac0) and film treated with acetic acid (Ac10) after 30 min, 5 h and 24 h. In this way, the film treated with acetic acid present the same absorption capacity than the alginate hydrogel.

Similar water sorption results have been reported for alginate films crosslinked with divalent cations of calcium and cobalt [2].

### 3.3. Toxicological Study

The results of the toxicity assays of the hydrogel extracts in the presence of human keratinocyte HaCaT cells are shown in Figure 3.

Extracts of an Ac0 sample showed no statistically significant changes in cell viability (%) to those of the positive control, although Ac10 showed statistically significant reductions in cell viability (%). However, the cell viability of extract of Ac10 was greater than 70%, indicating that the samples were not cytotoxic, according to ISO-10993. These films were thus biocompatible in the presence of human keratinocyte cells.

Serda et al. demonstrated the non-cytotoxicity of a low concentration of acetic acid in murine fibroblast [38]. The effects of gelatine fibres with acetic acid in human foreskin fibroblasts and human embryonic kidney cells showed that gelatine fibres with 25% acetic acid possessed a cell viability greater than 80% [39], which agreed with those obtained in our study.

### 3.4. Proliferation Assay

The proliferative activities of films in the keratinocyte cell line were studied in samples Ac0 and Ac10 to avoid raising toxicity by increasing the treatment time to 72 h and analyse whether Ac was able to induce cell proliferation (Figure 4).

After 72 h of exposure of the cells to the film extracts, the samples showed no statistically significant changes in cell viability compared with the control. The Ac10 film extracts did not induce proliferation in human keratinocyte HaCaT cells because the cell viability was much lower than the proliferation control (growth factor).

### 3.5. Anticancer Study

Melanoma cells (B16) were treated with serial dilutions of the Ac0 and Ac10 extract samples for 24 h (Figure 5a,b, respectively). Untreated film (Ac0) showed a similar mean than the control, as expected, whereas the diluted 1/10 of the extract of Ac10 showed significant differences with the control. The cell viability of the diluted 1/10 extract was, therefore, lower than 70%.

Colon cancer cells (HT-29) were treated with serial dilutions of the Ac0 and Ac10 extract samples for 24 h (Figure 5c,d, respectively). The untreated film (Ac0) and control present similar means, as expected, whereas the diluted 1/10 of the extract of Ac10 showed significant changes with respect to control, indicating that the cell viability of the diluted 1/10 extract was below 70%.

Kaminskyy et al. studied the anticancer activities of different compound derivatives of acetic acid against more than ten types of cancer lines and found that significant specific influence depends on the different cancer cell lines [29]. The presence of flavone in acetic acid is involved in anticancer activity against cancer cells [28]. This effect is due to the induction of cytokines producing a tumour-specific inflammatory responses [40]. In colon cancer cell lines, such as Caco-2 cells, exposure times of up to 6 h were enough to produce toxicity in the cells [25], whereas the production of ROS induced apoptosis in melanoma cells [26].

### 3.6. In Vivo Toxicity Tests

*Caenorhabditis elegans* is an ideal living model to perform in vivo toxicity studies as good as rat or mouse LD50 [41,42,43,44,45,46]. The acute toxicity results are shown in Figure 6a–c and the chronic toxicity results are shown in Figure 6d–f for survival rate, body length, and reproduction, respectively.

The acetic film extract showed no significant difference with respect to the positive control after 24 h of exposure (acute toxicity) (Figure 6a). After 72 h of exposure (chronic toxicity), the result was similar, so the Ac10 film extract showed a high survival rate (Figure 6d). The Ac0 sample extract showed a survival rate of more than 70%, indicating non-toxicity. Body length was the second parameter analysed to determinate the toxicity of the acetic hydrogel extract. Growth was measured by body length in a photo taken under a microscope by Motic Images Plus 3.0 software. The results showed no significant difference between the Ac10 and Ac0 extracts and the control after 24 h (Figure 6b) and 72 h (Figure 6e) of exposure. Reproduction was the last parameter analysed to determinate the toxicity of the acetic hydrogel extracts, for which three nematodes were placed on a new plate and incubated for 48 h. The eggs were then counted under a microscope. After 24 h of exposure (Figure 6c), the results showed a similar number of eggs between the acetic film extracts and the control. After 72 h of exposure (Figure 6f), the Ac10 and Ac0 hydrogels showed presented a similar result to 24 h.

### 3.7. Double-Stranded RNA Extraction and Quantification

The quantified amount of RNA showed no statistically significant decrease between the control and after the virus had been in contact with the different samples (Figure 7).

Therefore, the antiviral activity of the films could be studied because both viral particles did not remain attached to the Ac0 and Ac10 films before the antiviral tests. These results were in good agreement with those previously obtained with antiviral films of alginate crosslinked with Ca^2+^ and Zn^2+^ cations [32].

### 3.8. Antiviral Test

#### 3.8.1. Enveloped Bacteriophage Φ6

Sonication and vortexing were performed to release all the viral particles from the samples. The results of the antiviral tests showed that the films treated with acetic acid (Ac10) possessed antiviral activities (Figure 8) after 30 min, 5 h, and 24 h of viral contact with bacteriophage Φ6.

After 24 h, 5 h, and 30 min of contact with the SARS-CoV-2 viral model, bacterial lawns were seen to grow on the plate with few plaques (Figure 7). In contrast, 5 min of viral contact was not long enough to achieve potent antiviral activity. The untreated film (Ac0) displayed the same behaviour as Ac10 since calcium alginate films are antiviral against enveloped viruses such as bacteriophage phi 6 and SARS-CoV-2 [1]. The bacteriophage Φ6 plaque-forming units per mL (PFU/mL) are shown after being in contact with the Ac0 and Ac10 and compared with the control in Figure 9.

After 24 h and 5 h of viral contact with the bacteriophage Φ6, the % inactivation of the virus was 100% in both samples. When the viral contact declined, after 30 min, the % inactivation was 94.02% for Ac0 and 99.97% for Ac10 (Table 2). The log reduction was higher in Ac10 than in Ac0 (4.39 in Ac10 and 1.32 in Ac0). However, after 5 min of viral contact the % inactivation of the virus in Ac10 was 60.20 (Figure 8 and Table 2).

#### 3.8.2. Non-Enveloped Bacteriophage MS2

The samples were sonicated and vortexed to release all the viral particles. The results of the antiviral tests showed that the films treated with acetic acid (Ac10) possessed antiviral activities after 30 min, 5 h, and 24 h of viral contact with the bacteriophage MS2 (Figure 10).

After 24 h and 5 h of contact between the film treated with acetic acid (Ac10) and bacteriophage MS2, bacterial lawns were seen to grow on the plate with few plaques (Figure 10), whereas, after 30 min and 5 min of viral contact, there was little or no antiviral effects, respectively. The untreated film (Ac0) showed no antiviral activity against bacteriophage MS2. The plaque-forming units per mL (PFU/mL) of bacteriophage MS2 are shown after being in contact with the Ac0 and Ac10 and compared with the control in Figure 11.

The % inactivation of the virus was ~0% in the Ac0 film after 5 min, 30 min, 5 h, and 24 h of viral contact with the bacteriophage MS2 (Table 3). However, the % inactivation of the virus was 95.95% in the Ac10 film after 24 h of viral contact with the bacteriophage MS2. After 5 h, 30 min, and 5 min of viral contact, the % inactivation was 80.59, 64.50, and ~0%, respectively, in the Ac10 film.

Since the times of Ancient Greece, vinegar has been used as a disinfectant [16], as has been confirmed in many studies [17,18]. Therefore, acetic acid is a good candidate to study antiviral activity. Fernandez-Romero et al. carried out a time-addition antiviral experiment and concluded that an early event in the virus replication cycle was affected in the Herpes simplex virus [47]. Acetic acid destroyed the viral envelope and inhibited the transmission of the enveloped virus because it was inactivated and separated their external glycoproteins [48]. In addition, derivates of acetic acid induced the production of interferon, inhibiting the replication of viruses like Influenza [27] and Herpes simplex [22]. This mechanism could be acting on the antiviral activity of the bacteriophage phi6 model of SARS-CoV-2 [23]. Recently, acetic acid showed a stronger effect with reduced binding of the spike protein to ACE2, the main SARS-CoV-2 cell receptor [30].

In non-enveloped virus, like bacteriophage MS2 and non-enveloped norovirus, the mechanism is not clearly understood, although acetic acid could produce physical interactions with the virus and carry out its neutralization [49].

## 4. Conclusions

A novel alginate film crosslinked with Ca^2+^ and treated with acetic acid exhibited biocompatibility in vitro in human keratinocyte cells and in vivo in the *Caenorhabditis elegans* model. This hydrogel showed 99.97% of viral inactivation of enveloped viruses such as bacteriophage phi 6 after just 30 min of viral contact and 95.95% of viral inactivation of non-enveloped viruses such as bacteriophage MS2 after 24 h of viral contact. The film also exhibited anticancer properties against melanoma and colon cancer cells. These results show that this hydrogel film is a promising potential broad-spectrum antiviral and anticancer material for several biomedical and pharmaceutical drug delivery applications.

## Figures and Tables

**Figure 1 biomedicines-11-02549-f001:**
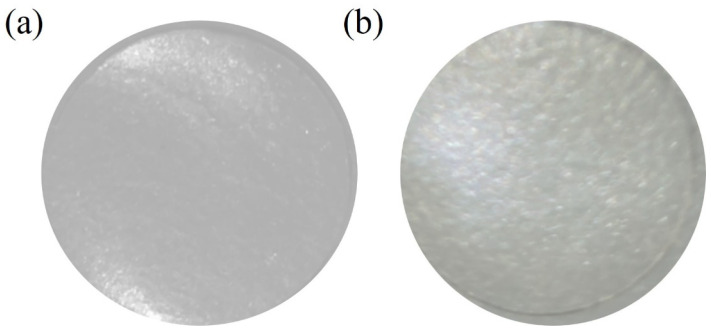
Macroscopic images of calcium alginate film containing acetic acid (**a**) and alginate film crosslinked with calcium (control film) (**b**).

**Figure 2 biomedicines-11-02549-f002:**
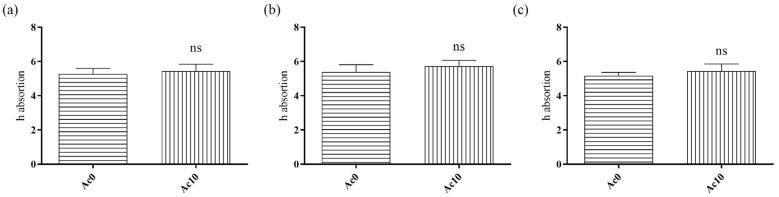
Absorption of liquid water ((m_hydrated film_ − m_dry film_)/m_dry film_) from untreated film (Ac0) and film treated with acetic acid (Ac10) after 30 min (**a**), 5 h (**b**), and 24 h (**c**) at 37 °C. The results of the statistical analysis of untreated film are indicated in the graph. ns: not significant.

**Figure 3 biomedicines-11-02549-f003:**
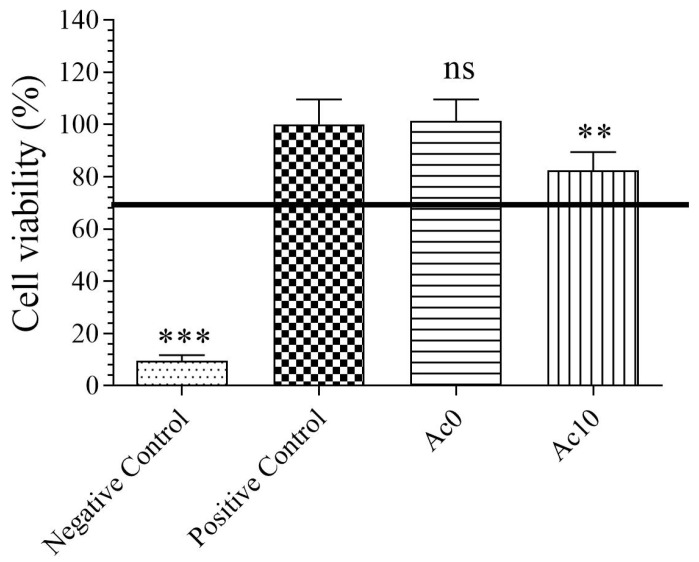
3-[4,5-dimethylthiazol-2-yl]-2,5-diphenyl tetrazolium bromide (MTT) cytotoxicity test of extracts acquired from untreated film (Ac0), film treated with acetic acid (Ac10), and positive and negative controls cultured with human keratinocyte HaCaT cells at 37 °C. *** *p* < 0.001; ** *p* < 0.01; and ns: not significant.

**Figure 4 biomedicines-11-02549-f004:**
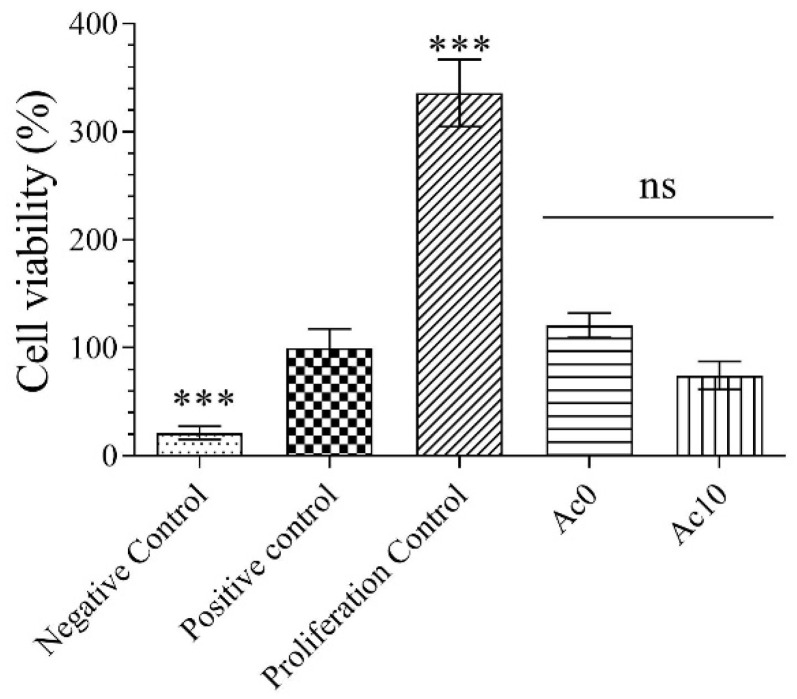
Proliferative activity of extracts of the untreated film (Ac0) and film treated with acetic acid (Ac10) in human keratinocyte HaCaT cells for 72 h. Data are expressed as the mean ± standard deviation of six replicates. The results of the statistical analysis with respect to control are indicated in the graph: *** *p* < 0.001; and ns: not significant.

**Figure 5 biomedicines-11-02549-f005:**
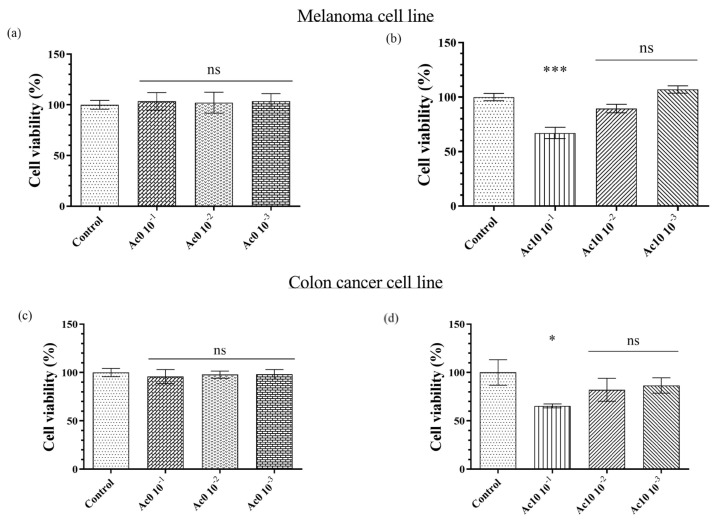
Anticancer activity of extracts of untreated film (Ac0) (**a**) and film treated with acetic acid (Ac10) (**b**) in melanoma (B16) for 24 h of exposure and anticancer activity of extracts of Ac0 (**c**) and Ac10 (**d**) in colon cancer cells (HT-29) for 24 h of exposure. The results of the extracts are shown for 10^−1^, 10^−2^, and 10^−3^ dilutions. Data are expressed as the mean ± standard deviation of six replicates. The results of the statistical analysis with respect to control are indicated in the graph: * *p* < 0.05; *** *p* < 0.001; and ns: not significant.

**Figure 6 biomedicines-11-02549-f006:**
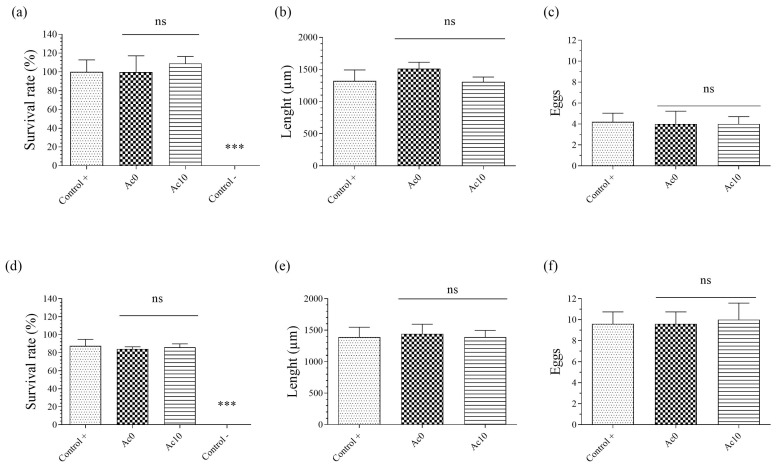
Measured parameters of the *Caenorhabditis elegans* in vivo toxicity model of exposure to extracts of untreated film (Ac0) and film treated with acetic acid (Ac10) after 24 h (acute toxicity): survival rate (**a**), body length (**b**), and reproduction (**c**); after 72 h (chronic toxicity): survival rate: (**d**) body length (**e**) and reproduction (**f**). Data are expressed as the mean ± standard deviation of six replicates (*n* = 6). The results of the statistical analysis with respect to positive control are indicated in the graph: *** *p* < 0.001, ns: not significant.

**Figure 7 biomedicines-11-02549-f007:**
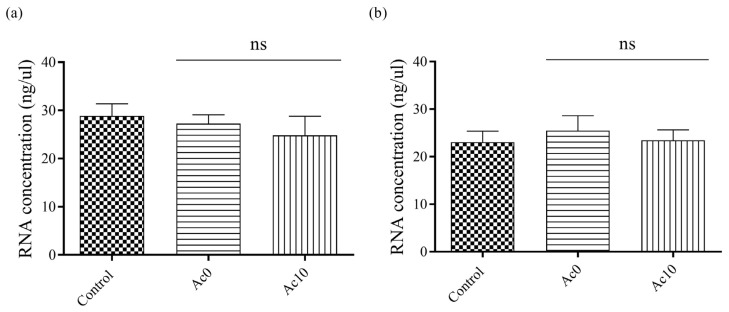
Quantified amount of RNA in ng/μL of bacteriophage phi6 (**a**) and bacteriophage MS2 (**b**) measured without being in contact with any samples (control) and the same amount of bacteriophage after being in contact with the Ac0 and Ac10 films for 24 h; ns, not significant.

**Figure 8 biomedicines-11-02549-f008:**
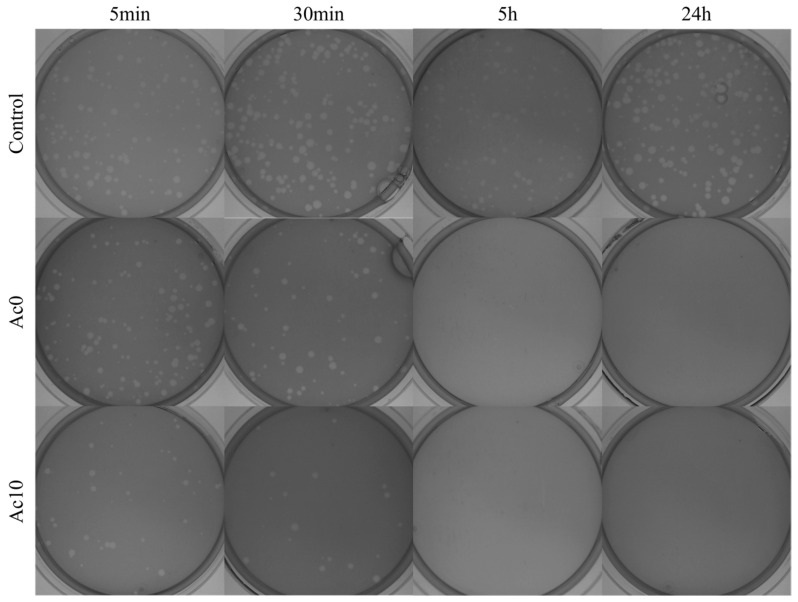
Loss of bacteriophage phi 6 viability measured by the double-layer method. Bacteriophage phi 6 titration images of undiluted samples for control, untreated film (Ac0), and film treated by acetic acid (Ac10) after 5 min, 30 min, 5 h and 24 h of viral contact.

**Figure 9 biomedicines-11-02549-f009:**
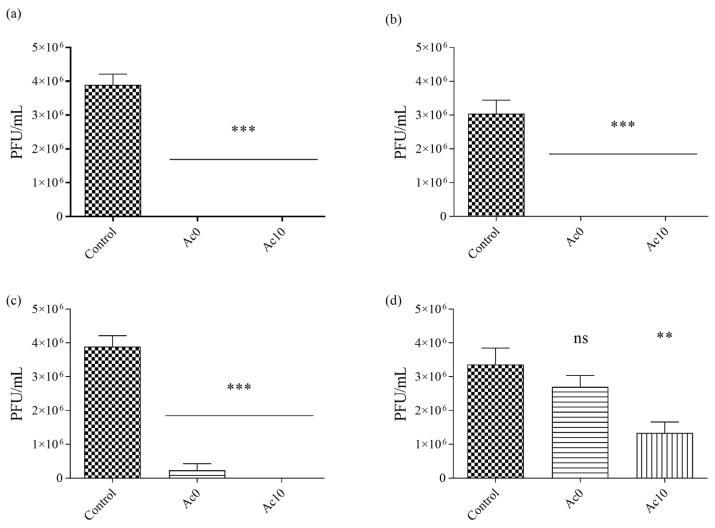
Reductions in infection titres of the phi 6 bacteriophage in plaque-forming units per mL (PFU/mL) measured by the double-layer method. Control, untreated film (Ac0), and film treated with acetic acid (Ac10) after 24 h (**a**), 5 h (**b**), 30 min (**c**), and 5 min (**d**) of viral contact. *** *p* < 0.001; ** *p* < 0.01; and ns, not significant.

**Figure 10 biomedicines-11-02549-f010:**
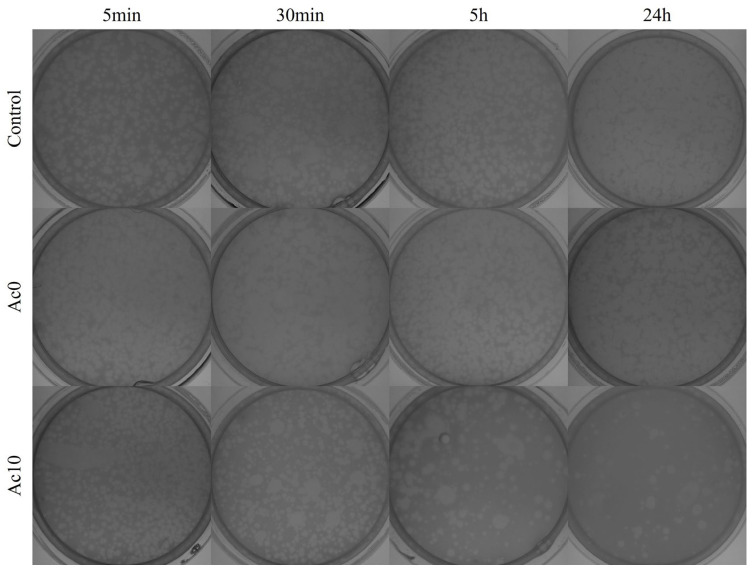
Loss of bacteriophage MS2 viability measured by the double-layer method. Bacteriophage MS2 titration images of undiluted samples for control, untreated film (Ac0), and film treated by acetic acid (Ac10) after 5 min, 30 min, 5 h, and 24 h of viral contact.

**Figure 11 biomedicines-11-02549-f011:**
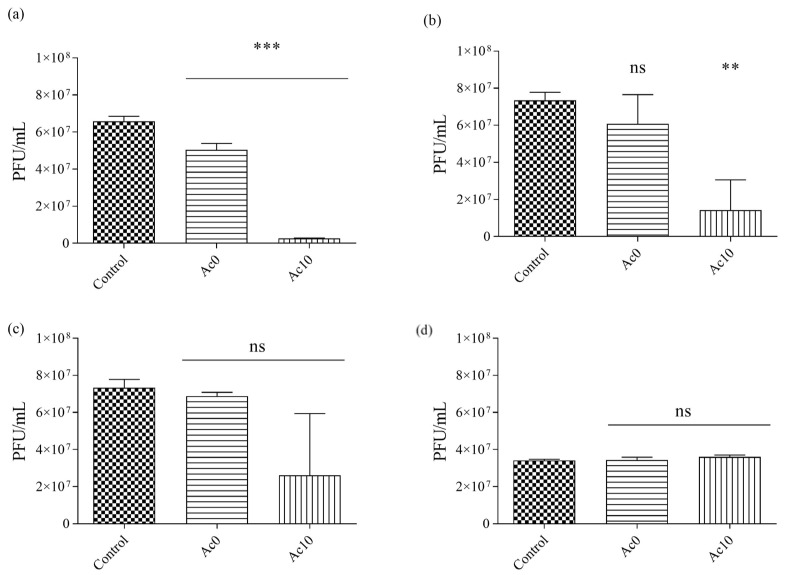
Reductions in infection titres of the MS2 bacteriophage in log (plaque-forming units per mL) (log(PFU/mL)) measured by the double-layer method. Control, untreated film (Ac0), film treated with acetic acid (Ac10) after 24 h (**a**), 5 h (**b**), 30 min (**c**) and 5 min (**d**) of viral contact. *** *p* < 0.001; ** *p* < 0.01; ns, not significant.

**Table 1 biomedicines-11-02549-t001:** Acetic acid present in the calcium alginate films (Ac0 and Ac10). Data (*n* = 6) indicated as mean ± standard deviation.

Sample Film	% Weight
Ac10	15.95 ± 1.45
Ac0	0.00 ± 0.00

**Table 2 biomedicines-11-02549-t002:** Infection titres obtained by the double-layer method for the antiviral assay performed, with bacteriophage Φ6 expressed as mean ± standard deviation, percentage of viral inactivation, and log (PFU/mL) reduction with respect to control and after contact with the untreated film (Ac0), and the film treated with acetic acid (Ac10) for 5 min, 30 min, 5 h, and 24 h.

		Control	Ac0	Ac10
5 min	PFU/mL	3.37 × 10^6^ ± 4.80 × 10^5^	2.70 × 10^6^ ± 3.34 × 10^5^	1.34 × 10^6^ ± 3.22 × 10^5^
	log reduction	-	0.10	0.41
	% inactivation virus	-	≈0	60.20
30 min	PFU/mL	3.89 × 10^6^ ± 3.23 × 10^5^	2.33 × 10^5^ ± 1.94 × 10^5^	1.33 × 10^3^ ± 1.15 × 10^3^
	log reduction	-	1.32	4.39
	% inactivation virus	-	94.02	99.97
5 h	PFU/mL	3.04 × 10^6^ ± 3.99 × 10^5^	0.00 ± 0.00	2.00 × 10^3^ ± 3.46 × 10^3^
	log reduction	-	6.48	5.22
	% inactivation virus	-	100.00	99.93
24 h	PFU/mL	3.89 × 10^6^ ± 3.23 × 10^5^	0.00 ± 0.00	0.00 ± 0.00
	log reduction	-	6.59	6.59
	% inactivation virus	-	100.00	100.00

**Table 3 biomedicines-11-02549-t003:** Infection titres obtained by the double-layer method for the antiviral assay performed on bacteriophage MS2 expressed as mean ± standard deviation, percentage of viral inactivation, and log(PFU/mL) reduction with respect to control after being in contact with the untreated film (Ac0) and the film treated with acetic acid (Ac10) for 5 min, 30 min, 5 h, and 24 h.

		Control	Ac0	Ac10
5 min	PFU/mL	3.41 × 10^7^ ± 6.93 × 10^5^	3.43 × 10^7^ ± 1.53 × 10^6^	3.60 × 10^7^ ± 1.00 × 10^6^
	log reduction	-	0	0
	% inactivation virus	-	0	0
30 min	PFU/mL	7.34 × 10^7^ ± 4.40 × 10^6^	6.88 × 10^7^ ± 2.11 × 10^6^	2.61 × 10^7^ ± 3.34 × 10^7^
	log reduction	-	0.03	0.71
	% inactivation virus	-	≈0	64.50
5 h	PFU/mL	7.34 × 10^7^ ± 4.40 × 10^6^	6.08 × 10^7^ ± 1.57 × 10^7^	1.42 × 10^7^ ± 1.64 × 10^7^
	log reduction	-	0.09	0.91
	% inactivation virus	-	≈0	80.59
24 h	PFU/mL	6.59 × 10^7^ ± 2.72 × 10^6^	5.04 × 10^7^ ± 3.47 × 10^6^	2.67 × 10^6^ ± 2.30 × 10^5^
	log reduction	-	0.12	1.39
	% inactivation virus	-	≈0	95.95

## Data Availability

Data is contained within the article.

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
