# Peer review of "Biocompatible Alginate Hydrogel Film Containing Acetic Acid Manifests Broad-Spectrum Antiviral and Anticancer Activities"

_biomedicines, 2023, doi:10.3390/biomedicines11092549_

Round 1
Reviewer 1 Report
In the present work, the authors reported the preparation of alginate hydrogel film crosslinked with Ca2+ and treated with acetic acid. This film exhibited prominent biological properties, including biocompatibility (in vitro in human keratinocyte cells and in vivo in the Caenorhabditis elegans), antiviral activity (against bacteriophage phi 6 and bacteriophage MS2) and anticancer properties (against melanoma and colon cancer cells). There are many issues in the manuscript. I would not recommend this work for publication unless the authors address these issues and make careful revisions.
Here are several issues:
1. The Abstract should be modified as its writing style is not suitable. It should be a paragraph instead of mentioning points one by one, such as background, methods, results, and conclusion.
2. The content of the Introduction is poor. The subject of this work is the modification of the alginate hydrogel film (acetic acid treatment), not only the acetic acid. This section should provide a review of the development of this film in the literature, including preparation, applications, properties, advantages, and limitations, which are the basis to emphasize the need to carry out the modification.
3. In Section 2.3, a summary of the measurement procedure should be provided for readers to follow the work, though this measurement has been done in the author’s previous work. The same thing should be done for Section 2.7.
4. Some photos of the prepared film should be provided for better imagination.
5. More characterizations (e.g., FTIR, SEM, etc) should be provided.
6. The discussion is poor in Sections 3.1, 3.2 and 3.7, with just a sentence. This is a scientific article, not an analytical report. A detailed discussion, together with explanations supported by references for each result, should be provided.
7. The sentence with the format “…showed no significant differences” was repeated many times in this manuscript. Authors should find other forms for discussion to avoid repetition.
8. In Figures 7 and 8, it is hard to observe bacterial lawns due to the poor contrast of images resulting from the blue background. These photos should be modified.
9. Why did the authors only work with one concentration of acetic acid? Do the biological properties of the prepared film change with increasing/decreasing acetic acid concentration? This should be investigated.
10. Acetic acid is soluble in various solvents such as water, alcohol, etc. Therefore, it could be easily removed by washing using these solvents. The stability/reusability of the prepared fill should be investigated.
Several changes are needed. (please see comments)
Author Response
The rebuttal letter is attached.

Reviewer 2 Report
Please find the comments attached

The English language is understandable and needs minor correction but the entire manuscript has many typo errors.
Author Response
Rebuttal letter attached.

Reviewer 3 Report
The manuscript "Biocompatible Alginate Hydrogel Film Containing Acetic Acid Manifests Broad-Spectrum Antiviral and Anticancer Activity" by Alba Cano-Vicent et al. presents promising findings on the development of alginate hydrogel films crosslinked with calcium ions and treated with acetic acid for biomedical applications. The manuscript has several strengths, including novel findings on the film's antiviral and anticancer properties. However, there are several areas that could be improved to enhance the manuscript's quality and suitability for publication.
The abstract provides a brief overview of the research but could be improved by reformatting it into a single cohesive paragraph that summarizes the study's objectives, methodologies, key results, and implications.
The introduction lacks depth and context. It would be beneficial to provide a comprehensive literature review on alginate hydrogel films, highlighting their preparation, applications, properties, advantages, and limitations. This contextual foundation would help to better emphasize the importance of the modification discussed in the manuscript. Author should more add some recent papers
1. J. Mater. Chem. C, 2022, 10, 12652–12679
2. Nanomaterials 2021, 11, 3009. https://doi.org/10.3390/nano11113009
3. DaltonTrans.,2020,49,8672–8683.
The measurement procedures should be briefly summarized to aid readers' understanding, even if they were previously detailed in the authors' earlier work.
Visual aids, particularly images of the acetic acid-treated alginate hydrogel film, would enhance readers' comprehension. Additionally, the clarity of most of the images could be improved.
The manuscript lacks crucial characterizations that probe the chemical bonding and surface morphology of the film, which are fundamental for understanding the film's structure. Integrating these analyses would bolster the study's scientific rigor.
The discussion in the results section could be significantly expanded. For example, in section 3.1, no information is provided on the gravimetric analysis. I suggest that the authors move beyond a single sentence to provide detailed insights and references for each result.
Considering a range of acetic acid concentrations would provide a more comprehensive understanding of its impact on the film's properties.
The manuscript should explore the stability and reusability of the prepared film, given acetic acid's solubility. Investigating the film's resilience to washing with various solvents would enhance its practical relevance.
Overall, the manuscript holds promise in its exploration of the modified properties of alginate hydrogel films. Addressing the highlighted concerns, including reformatting the abstract, enhancing the introduction, providing clear methodologies, incorporating visual aids, including essential characterizations, and engaging in comprehensive discussions, would significantly enhance the manuscript's quality. By doing so, the manuscript could make a valuable contribution to the biomedical and pharmaceutical sectors. I encourage the authors to address these points and resubmit the revised manuscript for further consideration.
Minor editing of English language required in manuscript
Author Response
Rebuttal letter attached.

Round 2
Reviewer 1 Report
The authors have done a careful revision of the manuscript. I recommend publication of this work in its present form.
Author Response
Dear Esteemed Reviewer, many thanks for your time and effort for providing us this expert revision opportunity and we are glad that you have recommended our revised manuscript suitable for publication.
Reviewer 2 Report
Please find the attachment

The quality of the English language is acceptable.
Author Response
Reviewer 2
Comment 1: I thank the authors to have revised the manuscript and have provide a point to point response to the manuscript titled “Biocompatible alginate hydrogel film containing acetic acid manifests broad-spectrum antiviral and anticancer activity” by Alba Cano-Vicent et al. Although the authors have provided additional information supporting to their manuscript, but the reviewer is still not satisfied. Few points that directs the judgment includes.
Response 1: Thank you very much for your positive comment.
Comment 2: The protocol of the authors describes dissolving the crystals from MTT assay using DMSO, but the response of the author to the reviewer for point no 4 says they have performed MTS assay. This creates a doubt in the reviewer’s mind.
Response 2: We apologize for this mistake that we have corrected now in the new revised version of the manuscript. We wrote MTT growth assay in section 2.6.( Anticancer study) and it is MTS growth assay. MTS (3-(4,5-dimethylthiazol-2-yl)-5-(3-carboxymethoxyphenyl)-2-(4-sulfonyl)-2H-tetrazolium). The MTT assay was used for the cytotoxicity assays.
Comment 3: The revised manuscript is still not concrete and comprehensive to address the points/concern raised by the reviewer.
Response 3: Thank you very much for your comment. We have addressed all the reviewers comments as better as possible including new references and new text highlighted with the Microsoft Word function
Reviewer 3 Report
The authors have not addressed my commnets properly. Also authors have mentioned some changes which they have not done in the manuscript. So,the further revison are required and do the proper correction before the final publication in this journal.
Author Response
Reviewer 3
Comment 1: The authors have not addressed my commnets properly. Also authors have mentioned some changes which they have not done in the manuscript. So, the further revison are required and do the proper correction before the final publication in this journal.
Response 1: We would like to express our gratitude for your valuable feedback on our manuscript. We apologize for any misunderstandings or oversights in our previous responses. We take your comments seriously and are committed to addressing them comprehensively to improve the quality of our manuscript.
We understand the importance of clarity and transparency in the revision process. We have carefully revisit all your comments and compare them with our responses to ensure that all your suggestions and concerns are appropriately addressed to be best of our ability and resources and the same incorporated into the revised manuscript.
We hope that this final version is accepted for publication.
Round 3
Reviewer 3 Report
The authors have made all the suggested observations and the paper can be accepted for publication.